



# Cloud probability-based estimation of black-sky surface albedo from AVHRR data

Terhikki Manninen[1], Emmihenna Jääskeläinen[1], Niilo Siljamo[1], Aku Riihelä[1], Karl-Göran Karlsson[2]

[1]Meteorological research, Finnish Meteorological Institute, Helsinki, FI-00101, Finland
[2]Atmospheric Remote Sensing Unit, Research Department, Swedish Meteorological and Hydrological Institute, Norrköping, SE-60176 Norrköping, Sweden

*Correspondence to*: Terhikki Manninen (terhikki.manninen@fmi.fi)

**Abstract.** Cloud cover constitutes a major challenge for the surface albedo estimation using Advanced Very High Resolution Radiometer AVHRR data for all possible conditions of cloud fraction and cloud type on any land cover type and solar zenith angle. Cloud masking has been the traditional way to estimate surface albedo from individual satellite images. Another approach to tackle cloudy conditions is presented in this study. Cloudy broadband albedo distributions were simulated first for theoretical cloud distributions and then using global cloud probability (*CP*) data of one month. A weighted mean approach based on the *CP* values was shown to produce very high accuracy black-sky surface albedo estimates for simulated data. The 90% quantile for the error was 1.1% (in absolute albedo percentage) and for the relative error it was 2.2%. AVHRR based and *in situ* albedo distributions were in line with each other and also the monthly mean values were consistent. Comparison with binary cloud masking indicated that the developed method improves cloud contamination removal.

## 1.   Introduction

The surface albedo is one of the key indicators of climate change (GCOS, 2016). Accurate solar and atmospheric radiation measurements are carried out practically continuously at fixed stations in contrasting climatic zones within the Baseline Surface Radiation Network (BSRN) project (König-Langlo et el., 2013; Driemel et al., 2018) of the World Climate Research Programme (WCRP). Remote sensing is the only reasonable alternative for augmenting regional surface albedo estimates globally. EUMETSAT provides the climate community with satellite based surface albedo products in the project Satellite

Application Facility on Climate Monitoring (CM SAF), which is part of the EUMETSAT Applications Ground Segment (Schulz et al., 2009). The CLARA (CM SAF cLoud, Albedo and surface Radiation dataset from AVHRR data) record contains cloud properties, surface albedo and surface radiation parameters derived from the AVHRR sensor onboard polar orbiting NOAA and METOP satellites. The CLARA-A2 (second edition) covered the years 1982-2015 and the next edition, A3, will cover the years 1979 – 2020.

The determination of surface black-sky albedo (Lucht et al. 2000, Román et al., 2010) from satellite data is usually carried out after first applying a cloud masking procedure. Thus, the accuracy of the cloud mask is really crucial to the albedo product. In





spite of augmenting information of global reanalysis data from ERA-5 (Hersbach et al., 2020), land mask and topography data, the demand of high accuracy pixelwise cloud masking of AVHRR images in all possible cloud fraction and type situations is extremely challenging, especially for the oldest satellites, due to lack of one of the two split-window infrared channels at 12 micron wavelength, and the high and variable noise levels in the 3.7 micron channel. Earlier comparisons of AVHRR cloud

masks with Moderate Resolution Imaging Spectroradiometer (MODIS) based cloud masks estimated that 1 – 3 % of the nominally clear local area coverage AVHRR data are cloud contaminated (Heidinger et al., 2002). More recent studies making use of high sensitivity lidar measurements from the Cloud-Aerosol Lidar and Infrared Pathfinder Satellite Observation (CALIPSO) mission indicate that the fraction of missed clouds is significantly higher and may even exceed 10 % in some geographical regions (Karlsson et al., 2017 and Karlsson and Håkansson, 2018). One factor that explains this difference to

earlier studies is that the CALIPSO-CALIOP lidar is able to observe also very thin clouds which are truly non-visible to AVHRR data. However, for the really large deviations also other cloudy vs clear non-separability issues become important. In particular, the presence of snow- and ice-covered ground and strong near-surface temperature inversions at high latitudes leads to a larger fraction of undetected clouds in AVHRR data. Using such data would then result in an extra error of similar size on the surface albedo values. Actually one is inclined to fear that even larger cloud masking errors will appear in difficult

topographies with snow cover and low sun elevation angles, which are common in Northern Europe.

Another approach to tackle cloudy conditions has been developed in this study. It appears that clear (or almost clear) sky and completely cloudy sky situations are much more frequent than the intermediate conditions (Manninen et al., 2004). Due to the great variation of cloud properties also the cloud albedo varies considerably. The temporally more slowly varying surface

albedo is thus expected to dominate the broadband albedo distribution of non-cloudmasked AVHRR data of one month in a grid box with a grid resolution of 0.25 x 0.25 degrees. The CLARA-A3 data record of CM SAF will provide the cloud probability as a new product. Here a method for estimating the surface albedo from the cloudy albedo distribution using the cloud probability values is presented. Theoretical simulations provide the basis for formulas used for estimating the cloud-free albedo distribution peak value without the need to construct the distribution itself, thus shrinking markedly the need of

computer resources. The results are compared with in situ measurements at snow-free and snow-covered test sites.

## 2.   Materials

### 2.1.   *In situ* albedo data

To verify the satellite-based albedo estimates, in situ surface albedo measurements were obtained from a selection of sites in the Baseline Surface Radiation Network (BSRN; Driemel et al., 2018). The sites Desert Rock (DRA), Southern Great Plains

(E13), Payerne (PAY), Fort Peck (FPE), Cabauw (CAB), Syowa (SYO), and Neumayer (GVN) were selected for their combination of albedo measurement availability with acceptable spatial representativeness of the site's measurement with respect to the albedo of the surrounding area, an important aspect for the point-to-pixel comparisons of satellite observations





with in situ measurements. The BSRN measurements are quality-monitored and the instruments regularly maintained, ensuring good quality as a reference dataset.

Additionally, data from the Summit Camp site of the Greenland Climate Network (Steffen et al., 1996) was used to add
coverage over ice sheet snow cover; the Summit site is often used as a snow albedo validation site for satellite studies due to the relatively low heterogeneity of the surface albedo in the area around the site.

### 2.2. AVHRR data

#### 2.2.1. FDR

The used AVHRR radiance data record is defined by applying the PyGAC preprocessing tool (Devasthale et al., 2017,
https://pygac.readthedocs.io/en/develop/) to the original AVHRR L1b data record hosted by NOAA. For the visible AVHRR channels PyGAC is using an updated calibration method originally formulated by Hedinger et al. (2010). Applicable calibration coefficients are described by the PyGAC documentation and they are based on the NOAA PATMOS-x calibration information published at https://cimss.ssec.wisc.edu/patmosx/avhrr_cal.html with its latest update in 2017. This data record has still not full Fundamental Climate Data Record (FCDR) status since the infrared channel radiances are not fully intercalibrated in the
same way as the visible channels. Consequently, the entire AVHRR data record will be published by EUMETSAT in 2022 as being an Fundamental Data Record (FDR) while a data record with full FCDR status is planned for release in 2026.

#### 2.2.2. Atmospheric correction

To achieve black-sky surface albedo (SAL) from the Top-Of-Atmosphere (TOA) reflectances, the atmospheric effects need to be removed. In processing of CLARA-A3 SAL this is done using the Simplified Method for Atmospheric Corrections (SMAC,
Rahman, H. and Dedieu, G., 1994) algorithm. The SMAC algorithm reduces the TOA reflectances to surface reflectances. In addition to TOA reflectances (as an output from Polar Platform Systems (PPS) pre-processing step), the SMAC algorithm needs other atmospheric input parameters (ozone content, surface pressure, total column water vapour content and aerosol optical depth (AOD) at 550 nm). For CLARA-A3 SAL, surface pressure, ozone content, and water vapour content are derived from ECWMF ERA5 global reanalysis data. The AOD at 550 nm are from the AOD time series by Jääskeläinen et al. (2017).
It is based on the Total Ozone Mapping Spectrometer (TOMS) and Ozone monitoring instrument (OMI) Aerosol Index (AI) data. Only AOD smaller than unity are used for SAL retrieval. For sea and permanent ice areas the constant AOD value 0.05 is used.

#### 2.2.3. Cloud probabilities

The new surface albedo retrieval approach makes use of some recent cloud masking developments taking place in the
EUMETSAT Nowcasting Satellite Application Facility (NWC SAF) project. The NWC SAF cloud processing package PPS



(Polar Platform System) has for many years provided cloud masks based on an original multispectral thresholding algorithm first described by Dybbroe et al., (2005). However, the latest version of PPS (denoted PPS version 2018) has added a complementary cloud masking method capable of providing cloud probabilities instead of fixed binary cloud masks as output. This product, denoted CMA-prob, is based on Bayesian retrieval theory and a first prototype method was described by Karlsson

et al. (2015). A substantially upgraded version, applied to both AVHRR and Spinning Enhanced Visible and InfraRed Imager (SEVIRI) data, was presented in Karlsson et a. (2020) and is now officially added to PPS version 2018. Results from this particular CMA-prob version have been utilised in this study.

The original cloud probability $CP$ values of CMA-prob per orbit in Global Area Coverage GAC resolution (~5 km) globally

for June 2012 were used as the starting point. Since the Surface Albedo Product (SAL) is delivered in a global grid of 1440 x 720 pixels, the pixelwise monthly distributions of $CP$ values were generated in that resolution. $CP$ values for cases where the solar zenith angle exceeds 70° were discarded for consistency with the same constraint in albedo calculations. Pixelwise distributions of $CP$ (altogether 31020) were calculated for every 5[th] pixel for the whole area, which naturally covered more the Northern Hemisphere due to illumination requirements. Examples of them are shown in Figure 1 for the 10 largest pixelwise

sets and the mean of all distributions. Obviously, very small cloud probability is more common than about 20%. The number or individual $CP$ values per distribution was on the average 1777, the 80% variation range being 203 – 2291. The number of CP values in the 10 largest distributions varied in the range 4064 – 4327.

The satellite based $CP$ values provided by the PPS software are used in Section 3.1.2 as the basis for simulations of the effect

of cloud fraction on the surface albedo. The $CP$ is taken to statistically represent the cloud fraction and only values smaller than 20% were used in this study, in order to achieve high accuracy for the surface albedo estimate. As the data mass even for every 5[th] pixel is quite large (620400 individual $CP$ values), the data was still reduced for the simulations the following way. First the ten largest distributions were taken, because they are statistically representative. Then every 50[th] set in decreasing order of the number of points in the distribution. Altogether 612 $CP$ distributions was used in the simulations. The reason to

use also very small distributions (the smallest set had only 14 $CP$ values) in the simulations was that such cases appear also, when deriving monthly albedo means using satellite data. For this data, the cloud probability did not mostly correlate with the solar zenith or azimuth angle. Hence, the simulations can be carried out combining any $CP$ values to any surface albedo values without paying attention to the solar angles. Further on, as the SAL product currently is not normalized to any specific solar zenith angle, the results of Section 3.1 can be applied to SAL processing without further consideration of solar angle effects

on the results.



## 3. Methods

### 3.1. Simulation of cloudy surface albedo distributions

#### 3.1.1. Theoretical cloud distributions

The cloud fraction varies between the two extremes, zero and unity, with varying weather conditions. When estimating the cloud fraction distribution over the entire globe in a very coarse spatial resolution, however, it is possible that the extreme values are not achieved at all. The ultimate limit is the planetary cloudiness, which is on the average about 66 % according to the latest report from the Global Energy and Water Exchanges GEWEX cloud assessment study (Stubenrauch et al 2021), the annual variation being about ± 5% (Karlsson and Devasthale, 2018). On the other hand, in very high spatial resolution the cloud fraction is typically clearly dominated by the extreme values, like ceilometer observations show (Manninen et al., 2004). Although the cloud probability estimation is complicated various kinds of uncertainties, the observed cloud fractions based on AVHRR data showed a U-curve resembling distribution both in original 1.1 km (Manninen et al., 2004) and the GAC resolution of ~5 km (Figure 1). Thus, it is more common to have completely cloud-free and completely cloudy pixels, but all intermediate values are also possible. A functional dependence adequate for fitting the observed cloud fraction curves seems to be (Manninen et al., 2004)

$$f_k(k) = exp(-ck) + b \, exp(-c(100 - k)) \tag{1}$$

where $k$ is the cloud cover percentage and $b$ and $c$ are parameters depending also on the spatial resolution.

The sun elevation dominates the diurnal variation of surface black-sky albedo (Briegleb and Ramanathan, 1982; Briegleb et al, 1986; Yang et al., 2008; Manninen et al., 2020). The diurnal albedo distribution is in snow-free areas almost symmetric, when the surface albedo is normalized with respect to midday, although the albedo is typically slightly lower in the morning than in the afternoon for the same solar zenith angle due to the presence of dew (Mayor et al., 1996). For snow cover during the melting season the albedo tends to be almost linearly decreasing during the day (Pirazzini 2004; Manninen et al., 2020; Manninen et al., 2021). In addition, the seasonal variation within one month may cause slight skewness in the albedo distribution.

The albedo of the surface and clouds should dominate the albedo distribution, because perfectly cloudy and perfectly clear skies are much more common than intermediate cloudiness. As long as the land use class does not change, the snow-free surface albedo has typically only moderate seasonal variation, but the albedo of clouds varies in a wide range with varying cloud type (Brisson et al., 1999). Therefore the monthly albedo distribution of a snow-free surface constitutes usually only one distinct peak, which is located roughly at the surface albedo value. This is the case also for snow cover in midwinter conditions, but during the melting season the distribution is much broader (Manninen et al., 2019). However, a Gaussian monthly albedo distribution $f(\alpha)$ is a reasonable approximation for both snow-free and snow-covered surfaces, i.e.



$$f(\alpha) = \int_{x=0}^{100} exp\left(-\frac{(x-\bar{x})^2}{2\sigma_x^2}\right) dx \tag{2}$$

where $\bar{x}$ is the surface albedo average, $\sigma_x$ is the standard deviation of the surface albedo distribution and $\alpha$ denotes the albedo

variable. The monthly albedo distribution observed by optical satellite radiometers can be described as a combination of the surface albedo distribution, the cloud coverage distribution and cloud shadow distribution (Manninen et al., 2004). The surface albedo distribution normalized with respect to midday is typically reasonably close to Gaussian distribution. The cloud albedo distribution can also be assumed Gaussian, although the standard deviation may be so large, that the result is essentially the same as for uniform distribution. No actual distribution shape is motivated for shadows, because their existence requires several

conditions to apply simultaneously: 1) the pixel in question must be clear, 2) there must be a cloud close enough in the neighbourhood 3) the Sun elevation and azimuth angles must be such that the Sun is on the same line with the pixel in question and the cloud casting the shadow. Since slightly shadowed pixels are more probable than completely shadowed pixels an exponentially decaying distribution was assumed for shadows according to

$$f_p(p) = \exp(-10p) \tag{3}$$

where $p$ is a uniform random variable in the range [0, 1]. Then the probability density function (PDF) $f_s$ of the possibly shaded pixels is defined as an integral of the product of the individual PDFs of the shadow $p$ and the surface albedo value $x$. The Kronecker delta function ($\delta$) is included in the integral to restrict the integration to possible combinations of $\alpha$, $p$ and $x$ and to

include only cases with the cloud fraction $k = 0$ so that (Manninen et al., 2004) the probability density function $f_s$ for possibly shaded cloud-free pixels is

$$f_s(\alpha) = \int_{x=0}^{100} \int_{p=0}^{1} exp\left(-\frac{(x-\bar{x})^2}{2\sigma_x^2}\right) exp(-10p) \ \delta\left(\alpha - x\ (1-p) + \frac{x}{2}p\right) \ \delta(k) \ dxdp \tag{4}$$

assuming that the lowest albedo value that will be caused by shadowing is half of the true value. This assumption was based on empirical observations using AVHRR data (Manninen et al., 2004).

The theoretical monthly albedo probability density function of cloudy pixels $f_c(\alpha)$ is likewise defined as an integral of the product of the individual PDFs of the cloud fraction $k$ (given here in percentage), cloud albedo value $y$, and the surface albedo value $x$.

value $x$. Again the Kronecker delta function is included in the integral to restrict the integration to possible combinations of $\alpha$, $k$, $x$ and $y$ so that





$$f_c(\alpha) = \int_{x=0}^{100} \int_{y=0}^{100} \int_{k=0}^{100} exp\left(-\frac{(x-\bar{x})^2}{2\sigma_x^2}\right) exp\left(-\frac{(y-\bar{y})^2}{2\sigma_y^2}\right) \left(exp(-c\,k) + b\,exp\big(-c(100-k)\big)\right) \delta\left(\alpha - \frac{(100-k)x+k\,y}{100}\right) dxdydk$$

(5)

where $\bar{y}$ is the cloud albedo average and $\sigma_y$ is the standard deviation of the cloud albedo. The cloud albedo distribution is here

assumed to be Gaussian, but sometimes the standard deviation is so large, that the result is essentially the same as for a uniform

distribution. The total PDF $f_t(\alpha)$ covering all cases is

$$f_t(\alpha) = \begin{cases} f_s(\alpha), & k = 0 \\ f_c(\alpha), & k > 0 \end{cases}$$

(6)

The location of the local maxima (and minima) of the albedo distributions of Eqs. 4 and 5 correspond to albedo values $\alpha$ for

which $f_t{'}(\alpha) = 0$. Since the integrals can't be determined in closed form, no explicit relationship between the peaks of the PDF

of the true surface albedo $f(x)$ and the PDF of the total albedo $f_t(\alpha)$ can be derived. Thus the albedo PDFs are simulated

numerically.

For cloud-free cases the average surface albedo value of an experimental albedo distribution can be determined as the mean

of the upper and lower half height locations of the albedo distribution (Manninen et al., 2004). For the theoretical Gaussian

distribution this equals the mean value precisely. For cloudy cases the total albedo distribution is mostly not symmetric and

using the mean of the lower and upper half height albedo values results in overestimation of low surface albedo and

underestimation of high surface albedo mean values. It is not possible to derive a functional relationship in closed form between

the clear sky and cloudy albedo means even for theoretical distributions. In addition, the cloud fraction and type vary in large

ranges, so that nothing can be assumed concerning the shape of the cloudy albedo peak. It may be distinctly skewed or almost

symmetric. It may dominate the whole distribution or the background may be at the half height level of the peak. Therefore

robust parameters assuming nothing of the shape of the peak are sought for determining the mean surface albedo.

In a previous study the half-height width and the ¾ height width of the cloudy albedo distribution peak was shown to be

suitable for the true albedo value determination (Manninen et al., 2004). However, constructing distributions and determining

the peak widths is numerically a very slow process. This is not feasible when processing a long times series (~ 40 years)

globally (1440 x 720 pixels) even on monthly basis. Therefore, in this study we present a solution to estimate the surface

albedo peak value without the need to construct the distribution. It is derived by simulations of cloudy albedo distributions

based on observed statistics of the newly available cloud probability values (see Section 3.1.2).





### 3.1.2. Satellite based cloud distributions

The satellite based cloud probability data provided by the PPS software (Karlsson et al., 2020) were used as proxies for cloud fractions. The total albedo distributions were calculated separately for each pixelwise $CP$ distribution $f_{CP}(CP)$. Thus, the equations to use for satellite based versions of the cloud-free, but possibly shaded, pixel distribution $f_s$ and cloudy pixel

5    distribution $f_c$, $f_{Ss}$ and $f_{Sc}$ respectively, for each individual pixelwise cloud probability distribution are now

$$f_{Ss}(\alpha) = f_{CP}(0) \int_{x=0}^{100} exp\left(-\frac{(x-\bar{x})^2}{2\sigma_x^2}\right) \; exp(-10p) \; \delta\left(\alpha - x\,(1-p) + \frac{x}{2}p\right) \; dx \qquad (7)$$

$$f_{Sc}(\alpha) = \sum_{k=1}^{19} f_{CP}(k) \, exp(-d\,k) \int_{x=0}^{100} exp\left(-\frac{(x-\bar{x})^2}{2\sigma_x^2}\right) dx \qquad , \qquad (8)$$

where $k$ is the cloud probability discretized to integers in the range [1,19], because only $CP$ values smaller than 20% are allowed in order to achieve high estimation accuracy. As larger $CP$ values than that are not used in the analysis, it is sufficient to replace the term $exp\,(-c\,k) + b\,exp\,(-c\,(100 - k))$ of Eq. 5 by $exp(-d\,k)$. The parameter $d = 0.1$ is used to give even more weight for less cloudy albedo retrievals yet allowing some weight (0.135) also to the 20% cloud probability cases. The choice

of the value of $d$ is a compromise between theoretical accuracy and desire to avoid dominance of individual completely cloud-free retrievals. The reason not to integrate Eq. 7 over the parameter $p$ like in Eq. 4 is purely practical: the data mass is too large for that. Hence, just one random shadow value is taken into account per a $CP$. Each cloud probability distribution is combined with different values of the surface albedo distribution using random weights to improve the generalization of the results, as satellites observe samples of the surface albedo distribution in varying cloud conditions. The total albedo distribution $f_{St}(\alpha)$ is

then derived as a combination of those two alternatives as before using Eq. 6, where $f_t$, $f_s$ and $f_c$ are replaced by $f_{St}$, $f_{Ss}$ and $f_{Sc}$ respectively. The first estimate of the surface albedo mean $\bar{\alpha}$ is then obtained from $n$ individual albedo values as follows

$$\bar{\alpha} = \sum_{i=1}^{n} \alpha_i \, f_{St}(\alpha_i) / \sum_{i=1}^{n} f_{St}(\alpha_i) \qquad (9)$$

The monthly standard deviation, skewness and kurtosis are then calculated similarly using the total albedo distribution as weights. When a sufficient number of cloud-free pixels is present, this formula will give a good estimate for the surface albedo. However, if all pixels have 20% probability, the above formula will approach the albedo value corresponding to 20% cloud probability, not 0% cloud probability. Hence, an additional correction terms is applied to retrieve the final albedo estimate $\hat{\alpha}$ in the form

$$\hat{\alpha} = \bar{\alpha}\left(1 + c_1 \overline{CP} - \frac{c_2\,\overline{CP}}{\bar{\alpha}}\right) \qquad (10)$$





where $\overline{CP}$ is the monthly mean cloud probability of the $CP$ values within the range $[0\%, 20\%)$ and $c_1$ and $c_2$ parameters are determined empirically on the basis of the simulations. The assumed cloud albedo mean and standard deviation were 60% and 20%, respectively, and the calculations were made for Gaussian surface albedo with mean values 10%, 20%, 30%, 40%, 50%, 60%, 70% and 80% and with a standard deviation of 2%. The values of $c_1$ and $c_2$ producing the best fit of estimated albedos

to the true ones are given in Table 1. This formula adjusts the albedo estimate only, when $\overline{CP}$ exceeds zero. The standard deviation, skewness and kurtosis estimates based on $\bar{\alpha}$ are corrected similarly using the correction factor in brackets of Eq. 10, but with the dedicated parameter values of $c_1$ and $c_2$ given in Table 1.

### 3.2. Surface albedo retrieval algorithm

The surface albedo algorithm used in the Climate-SAF project starts with atmospheric correction carried out using the SMAC
method (Rahman and Dedieu, 1994: Proud et al, 2010). The next step is to determine the albedo values for the visible and infrared channels with the generally used formulas and coefficients for BRDF of various land use classes, which are taken from a land cover product (Roujean et al., 1992; Wu et al., 1995; Hansen et al., 2000). A topography correction is carried out in mountainous areas (Manninen et al., 2011). Finally, a broad band conversion is carried out (Liang, 2000; Liang et al., 2002). Currently, no solar zenith angle normalization is used, because at the time of the product development no generally applicable
formula existed for all surface types, including melting snow (Manninen et al., 2020). The previous SAL versions (Riihelä et al. 2013, Karlsson et al., 2017; Anttila et al. 2018) relied on cloud masking  applying the SAL algorithm only on nominated clear-sky pixels.

For the next release, CLARA-A3 SAL, the cloud probability values $CP$ provided by the PPS software (Karlsson et al., 2020)
will be available and the black-sky surface albedo (Lucht et al., 2000; Róman et al., 2010) retrieval will be based on pixels with cloud probability not exceeding 20 %. The albedo processing is first carried out as if all those pixels were completely cloud-free, i.e. the atmospheric correction for AOD, water vapour, air mass, and ozone is made. Then the monthly mean values $\bar{\alpha}$ are approximated similarly as in Section 3.1.2

$$\bar{\alpha} = \sum_{i=1}^{n} \alpha_i \exp(-d\ CP_i) / \sum_{i=1}^{n} \exp(-d\ CP_i) \qquad (11)$$

The theoretically motivated form of Eq. 10 for correcting $\bar{\alpha}$ turned out to result in slight black-sky albedo overestimation for large, especially sea ice, albedo values, when comparing to previous albedo time series. Since Eq. 11 is not a precise theoretical formula for deriving the cloud-free albedo using possibly cloudy data, but rather a practical statistical approach for its
estimation, it is understandable that a theoretical correction factor form may not be optimal either. Thus, finally an ordinary linear regression based empirical correction of $\bar{\alpha}$ was derived using the albedo simulations (Section 3.1.2). Hence, the final monthly mean albedo estimates $\hat{\alpha}$ were derived instead of Eq. 10 using the following formula





$$\hat{\alpha} = 1.0332\,\bar{\alpha} - \overline{CP}(-0.05600 + 0.007026\,\bar{\alpha}) \qquad\qquad . \tag{12}$$

The difference between $\hat{\alpha}$ and $\bar{\alpha}$ is rather small and consequently the standard deviation, skewness and kurtosis were still
corrected using Eq. 10.

## 4.   Results

### 4.1.  Simulated distributions

Albedo distributions were simulated for surfaces with Gaussian mean albedo values 10%, 20%, 30%, 40%, 50%, 60%, 70%
and 80% and a standard deviation of 2%. The Gaussian cloud albedo mean was taken to be 60% with a standard deviation of
20%. Examples of them are shown in Figure 2. For convenience all distributions are scaled so that the maximum equals unity
instead of using the common normalization of PDFs which would set the integral to unity and consequently cause varying
peak height. Obviously, for relatively low surface albedo values (such as those of vegetation) the clouds cause a tail at the high
end of the albedo distribution and for high albedo values (such as those of snow) a tail at the low end. For albedo values close
to the cloud albedo (such as sea ice albedo values) the distribution spreads both to low and high values. The bump at the cloud
albedo mean is more distinct for high large $b$ and $c$ of Eq. 5. Due to the larger standard deviation of the cloud albedo distribution
and the variation of the cloud probability, the surface albedo distribution peak still dominates the total distribution.

Albedo distributions were also derived using the empirical $CP$ pixelwise distributions and Eqs. 11 and 10. The results were
compared with the true values (Table 2). For the simulated Gaussian albedo distributions the obtained estimation accuracy is
very good: the mean absolute difference is 0.48% and the 90% quantile for the mean value is 1.1% (in absolute albedo
percentage). The relative mean albedo difference is 1.1% and the 90% quantile for the relative difference is 2.2%. But also
larger deviations appear: the maximum mean albedo error is 2.8% (in absolute percentage) and the relative mean albedo error
is 7.8%. The effect of the number of individual points in the simulation on the accuracy and relative accuracy of the albedo
estimation is shown in Figure 3. Naturally, large number of points increases the accuracy, but the effect is not dramatic. This
is important from the point of view of satellite images, because the number of individual points for a monthly mean may be
quite small in areas, where the sun elevation is typically small or the sky cloudy.

The reason for the very high accuracy is partly due to the assumed purely Gaussian surface albedo distributions that are
provided for the whole range [0%, 100%] with an increment of 1%. For satellite data the surface albedo distributions are patchy
and the sampling may be biased to certain solar zenith angle values. In addition, the distributions may deviate clearly from
Gaussian. Moreover, the number of individual satellite based albedo estimates per pixel may be much smaller than the 101



used in these simulations. Hence, it is not expected to achieve this high accuracy for the satellite based albedo estimates, but in principle this approach is capable of achieving very high albedo estimation accuracy.

### 4.2.  Satellite based distributions

Histograms of surface albedos in GAC resolution for one month were constructed using the new CLARA-A3 SAL algorithm for the chosen relatively homogeneous test sites Desert Rock (36.626°N, 116.018°W), Payerne (46.815°N, 6.944°E), Southern Great Plains (36.605°N, 97.485°W) and Greenland Summit (72.580°N, 38.500°W) with 1% bin width (Figure 4). The corresponding *in situ* albedo distributions are shown as well. Only those points are shown, for which both a satellite overpass and an *in situ* measurement were available within a 15 minute time window. Monthly means derived from simultaneous *in* situ and satellite based albedo values are given for several site in Table 3. Since the irradiance of the *in situ* measurements contains also contribution from the atmosphere, the comparison to a black-sky surface albedo estimate contains some inherent discrepancy, but for dark surfaces the *in situ* albedo values should be only slightly larger than the satellite based values. The difference increases with increasing AOD For bright targets, such as snow, the effect of the atmosphere reduces the measured surface albedo value. In addition, the satellite pixel diameter is about 25 km and the *in situ* measurements typically characterize footprints of some hundreds of square meters. Possible land cover inhomogeneity around the measurement site inevitably causes discrepancy between the satellite and *in situ* values (Riihelä et al., 2013). The difference between the albedo means shows slight increase with decreasing number of individual values behind the means and increasing distance between the mean of the satellite pixel locations and the measurement point.

In sea ice areas the variation of the surface albedo within one month may be large due to large amounts of open water and movement of the ice field. Examples of that are shown in Figure 5 for sites in Arctic Ocean of Alaska, Kara Sea and Laptev Sea. In 2009 there was in June still quite a lot of sea ice in all those three areas, whereas in June 2018, due to the climate change, all of those sites were relatively ice free, especially Laptev sea having a large area of open water (EUMETSAT, 2021). When the sea ice concentration varies markedly, the monthly mean albedo estimate is largely affected by the timing of days with small cloud probability.

### 5.  Discussion

This study demonstrates the use of cloud probability information for surface albedo retrieval. At the time of the study only one month of cloud probability data was available globally. In June the northern hemisphere is covered best, but highest southern hemisphere latitudes, mainly Antarctica are missing, because of the low solar zenith angle values. As the cloud cover varies seasonally, it would be desirable to update the parameter values of this study (Table 1 and the form of Eq.12) using global cloud probability data of one year. However, despite of the rather limited cloud probability statistics of this study, the achieved estimation accuracy of monthly means of albedo was satisfactory and the values were in line with the *in situ* measurements.





Typically the surface albedo of snow-free surfaces depends on the solar zenith angle so that the minimum is obtained at midday and the albedo is azimuthally symmetric (Briegleb and Ramanathan, 1982; Briegleb et al, 1986; Yang et al., 2008; Manninen et al., 2020). Also for snow outside the melting season the dependence is similar, unless the surroundings are very anisotropic.

If the whole diurnal albedo distribution were available using satellite data, it might be a good idea to take that into account in the simulations and deriving the parameter values to be used for albedo estimation (Table 1). However, a satellite based AVHRR instrument observes a site typically only once per day per pass and about at the same time on successive days. Hence, using a Gaussian albedo distribution as a basis for the simulations seemed reasonable.

The PPS software provides also a binary cloud mask (Karlsson et al., 2020), which can be used for surface albedo retrieval. Although the cloud mask and the $CP$ values are mostly strongly correlated, there are situations, when a pixel with $CP < 50\%$ is masked cloudy and a pixel with $CP > 50\%$ is masked cloud-free. This is because the underlying algorithms for cloud mask and cloud probability values are different. Distributions for cloud probability values below 50% corresponding to the cloud mask values zero (cloud-free) and unity (cloudy) are shown in Figure 6. Out of the total 35 million $CP$ values smaller than

50% the fraction 9.6% was masked cloudy. For the 27 million $CP$ values smaller than 20%, the fraction of cloudy classified pixels was 5.7%. Since the cloud fraction affects directly the top of atmosphere (TOA) reflectance, it also affects directly the surface albedo value. Hence, at tighter limit for cloudiness than 50% was considered preferable. The chosen limit $CP < 20\%$ is a compromise between the quality of TOA reflectance values and the number of pixels available for a monthly mean albedo retrieval. The reason for the pixels with $CP$ smaller than 50% classified as cloudy to be most common for the smallest $CP$

values is just that the smallest values dominate the lower end of the $CP$ distribution. It is indicated by the essentially linear increase from about zero to about 0.5 of the ratio of cloudy masked pixels and cloud-free pixels (Figure 6). This is exactly what one would expect a cloud mask to do.

For comparison to the approach of this study, surface albedo was estimated also using a standard cloud masking procedure.

Then the monthly mean albedo was directly the average value of cloud-free masked pixels, but also here we restricted the processing to cases with $CP < 20\%$. The results are shown in Figure 7. Obviously, using the cloud mask one would typically get slightly higher albedo values for snow-free areas than when using the cloud probability values and lower albedo values for snow-covered areas (Greenland Summit, Syowa and Neumayer). This is exactly how cloudiness would affect the albedo retrieval (Section 3.1.1) and supports the notion that the $CP$ based approach of this study can exclude the effect of cloud

contamination of the TOA reflectance values more effectively than plain cloud mask usage. In addition, the difference between the estimates of the two methods is typically largest for snow-covered areas, where cloud discrimination is very challenging, especially when the sun elevation is low (Karlsson and Håkansson, 2018).

The largest difference between the two approaches in Cabauw took place in November 2008, when there were only three points available matching the *in situ* measurement times, due to the low solar elevation, and the satellite-based albedo estimate varied in the range 20.6% - 43.3%. The largest value was masked cloud-free, but the *CP* value was 19.8% and the high reflectivity may as well be due to patchy snow or partial cloud contamination. Snow might also explain the somewhat larger fraction of

cloud masked pixels with very small *CP* values (Figure 6).

The CLARA-A3 SAL will be derived using the *CP* values instead of the binary cloud mask. The pentad means will be derived technically similarly as the monthly means using pentad distributions of *CP*. Future studies of the CLARA-A3 *CP* and cloud mask characteristics will show, whether it would be desirable to use both the cloud mask and the *CP* values as the basis for

SAL estimation. In addition, the parameter values to be used in Eqs. 10 and 12 would benefit from an updated analysis, using *CP* data for a whole year as input.

Since the surface albedo is directly related to the TOA reflectance value, the approach presented here for surface albedo estimation could be adapted also to estimating other reflectance-associated surface parameters instead of using the traditional

cloud masking, when a time window containing several images is of interest. Naturally, in general the reliability of the method increases with increasing number of points to be averaged.

## 6.    Conclusions

Cloud probability values to be provided by the CM SAF CLARA-A3 data record offer a good alternative to binary cloud masking for surface reflectance and albedo estimation, when the goal is not in studying individual images, but statistics within

a time window. Simple weighted averaging on the basis of the cloud probability values and a basic linear regression correction for biased no clear-sky events containing time windows provide good estimates for surface albedo.

**Acknowledgements**

This study was carried out in the project Satellite Application Facility on Climate Monitoring (CM SAF), financially supported by EUMETSAT. The authors wish to acknowledge the valuable support from the World Radiation Monitoring Center

(WRMC) in granting permission to use the BSRN data in this study. The Greenland Climate Network is acknowledged for the provision of data from Summit Station, Greenland. They also wish to thank other colleagues from FMI, and Climate-SAF project for co-operation during various parts of the work.

**Code availability**

No special code to deliver, only very basic calculations carried out using Mathematica.



**Data availability**

The cloud probability data used was a test set of a fundamental data record to be published by CM SAF.

**Author contribution**

5  TM, EJ and NS carried out the theoretical development. TM made the simulations and albedo quality analysis. AR provided *in situ* data and participated in the albedo quality analysis. KGK developed and provided the cloud probability data and related information used in the study. Everybody participated in writing the manuscript.

**Competing interests**

10  The authors declare that they have no conflict of interest.

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



**Appendix: Used parameter symbols**

| Symbol | Meaning |
|---|---|
| $c_1$ | Empirical coefficient used in final albedo estimate retrieval |
| $c_2$ | Empirical coefficient used in final albedo estimate retrieval |
| $CP$ | Cloud probability |
| $\overline{CP}$ | Monthly mean cloud probability |
| $d$ | Weight parameter for albedo monthly mean retrieval |
| $f(\alpha)$ | Albedo distribution |
| $f_c(\alpha)$ | Probability density function of albedo for cloudy pixels |
| $f_{CP}(CP)$ | Pixelwise cloud probability density function |
| $f_k(k)$ | Cloud distribution based on cloud fraction |
| $f_p(p)$ | Probability density function of shadows |
| $f_s(\alpha)$ | Probability density function of albedo for possibly shaded pixels |
| $f_{Sc}(\alpha)$ | Satellite based probability density function of albedo for cloudy pixels |
| $f_{Ss}(\alpha)$ | Satellite based probability density function of albedo for possibly shaded pixels |
| $f_{St}(\alpha)$ | Satellite based total probability density function of albedo |
| $f_t(\alpha)$ | Total probability density function of albedo |
| $k$ | Cloud fraction |
| $p$ | Probability of shadow |
| $\alpha$ | Albedo |
| $\bar{\alpha}$ | Monthly mean albedo, first estimate |
| $\hat{\alpha}$ | Monthly mean albedo, final estimate |
| $\beta$ | Kurtosis |
| $\gamma$ | Skewness |
| $\sigma$ | Standard deviation |



**Tables**

**Table 1. Parameter values of Eq. 10 for monthly mean, standard deviation, skewness and kurtosis of surface albedo.**

| Variable | $c_1$ | $c_2$ |
|---|---|---|
| Mean | 0.006343 | -0.1335 |
| Standard deviation | -0.0005595 | -0.04121 |
| Skewness | 0.008168 | 0.05647 |
| Kurtosis | 0.001205 | 0.1137 |





**Table 2. Simulated statistics for the absolute and relative differences of the estimated (^) and true values of albedo ($\alpha$), standard deviation ($\sigma$) skewness ($\gamma$) and kurtosis ($\beta$). The calculations were made for albedo values 10%, 20%, 30%, 40%, 50%, 60%, 70% and 80%. The albedo values are in the range 0 – 100%.**

| | $\lvert\hat{\alpha}-\alpha\rvert$ | $\dfrac{\lvert\hat{\alpha}-\alpha\rvert}{\alpha}$ | $\lvert\hat{\sigma}-\sigma\rvert$ | $\dfrac{\lvert\hat{\sigma}-\sigma\rvert}{\sigma}$ | $\lvert\hat{\gamma}-\gamma\rvert$ | $\dfrac{\lvert\hat{\gamma}-\gamma\rvert}{\gamma}$ | $\lvert\hat{\beta}-\beta\rvert$ | $\dfrac{\lvert\hat{\beta}-\beta\rvert}{\beta}$ |
|---|---|---|---|---|---|---|---|---|
| Mean | 0.48 | 0.011 | 0.0079 | 0.0076 | 0.038 | 0.0098 | 0.0033 | 0.018 |
| Median | 0.32 | 0.0089 | 0.0063 | 0.0041 | 0.020 | 0.0056 | 0.0018 | 0.010 |
| 90 % quantile | 1.1 | 0.022 | 0.019 | 0.018 | 0.09 | 0.022 | 0.0083 | 0.042 |
| Max | 2.8 | 0.078 | 0.037 | 0.058 | 0.30 | 0.071 | 0.035 | 0.124 |





**Table 3. Monthly mean black-sky surface albedo values based on AVHRR reflectance and *CP* values and the monthly means values of the corresponding times of *in situ* surface albedo measurements for several BSRN *in situ* sites (König-Langlo et el., 2013; Driemel et al., 2018): Desert Rock (Augustine, 2009a, 2009b, 2019a), Fort Peck (Augustine, 2009c, 2009d, 2019b), Payerne (Vuilleumier, 2010a, 2010b, 2019), Southern Great Plains (Long, 2009a, 2009b), Cabauw (Knap, 2018), Syowa (Yamanouchi, 2010) and Neumayer Station (König-Langlo, 2009). Date from the Greenland Summit *in situ* site is also included (Steffen et al., 1996). The number of observations included in the mean value are given as well as the mean distance of the satellite pixels from the *in situ* measurement mast.**

| Site | Location | Time | Number of observations | Mean distance [km] | In situ albedo [%] | AVHRR based black-sky albedo [%] |
|---|---|---|---|---|---|---|
| Desert Rock | 36.626°N, 116.018°W | November 2008 | 69 | 3.0 | 21.5 | 20.3 |
| | | April 2009 | 120 | 3.1 | 20.7 | 20.0 |
| | | July 2018 | 124 | 2.6 | 20.7 | 21.0 |
| Fort Peck | 48.31°N, 105.1°W | November 2008 | 19 | 3.1 | 18.8 | 15.8 |
| | | April 2009 | 90 | 2.9 | 17.3 | 15.2 |
| | | July 2018 | 177 | 2.7 | 16.2 | 17.4 |
| Payerne | 46.815°N, 6.944°E | November 2008 | 17 | 3.2 | 24.6 | 17.6 |
| | | April 2009 | 124 | 3.1 | 23.3 | 21.3 |
| | | July 2018 | 151 | 2.5 | 21.6 | 19.8 |
| Southern Great Plains | 36.605°N, 97.485°W | November 2008 | 65 | 2.0 | 20.9 | 19.3 |
| | | April 2009 | 77 | 2.9 | 20.2 | 20.2 |
| Cabauw | 51.971°N, 4.927°E | July 2018 | 171 | 2.9 | 23.0 | 19.7 |
| Syowa | 69.005°S, 39.589°E | November 2008 | 63 | 2.9 | 81.3 | 80.7 |
| Neumayer Station | 70.65°S, 8.25°W | November 2008 | 80 | 2.6 | 82.3 | 82.8 |
| Greenland Summit | 72.580°N, 38.500°W | April 2009 | 79 | 3.1 | 84.4 | 85.4 |



**Figure captions**

**Figure 1.** Ten largest pixelwise relative cloud probability distributions and the mean (dashed curve) of all *CP* distributions scaled with its maximum value. The solar zenith angle is restricted to not exceed 70°.

**Figure 2.** Examples of simulated cloudy albedo distributions for diverse values of parameters *b* (blue shades) and *c* (diverse values of dashing) in Eq. 5. The example surface albedo values are assumed to be Gaussian with mean values 20% (top left), 50% (top right) and 80% (top bottom) and a standard deviation of 2%. The surface albedo mean is shown as a red line and the distribution as a yellow curve. The parameter *p* in Eq. 4 had a random value in the range [0,1]. The example cloud albedo value is assumed to be Gaussian with a mean of 60% and standard deviation of 20%.

**Figure 3.** The relationship between the number of points in the cloud distribution and the simulated mean (left) and relative mean (right) albedo accuracy.

**Figure 4.** The albedo retrieval distributions at Desert Rock, Payerne and Southern Great Plains in April 2009 and Syowa in November 2008. The cloud probability values of the individual black-sky satellite based estimates are indicated by the colours. The monthly mean estimate is shown in red dashed line.

**Figure 5.** The albedo retrieval distributions at Arctic Ocean (73.370°N, 139.180°W), Kara Sea (2.680°N, 62.860°E) and Laptev Sea (75.320°N, 125.720°E) in April 2009 and July 2018. The cloud probability values of the individual black-sky satellite based estimates are indicated by the colours. The monthly mean estimate is shown in red dashed line.

**Figure 6.** The relative distributions of cloud probabilities for pixels masked cloud-free and cloudy provided by the PPS software for June 2012. The cloud-free masked pixel distribution is based on 26 million individual values and the cloudy masked pixel distribution on 1.6 million individual values. The ratio of the number of pixels classified as cloud to the number of pixels classified as cloud-free is shown as well.

**Figure 7.** The difference between the monthly mean surface albedo estimates derived using the cloud masking (CM) and cloud probability (CP) approaches for several Northern hemisphere test sites in July 1979, April 1981, November 2008, April 2009, July 2018 and April 2020. For Greenland Summit no satellite albedo data was available for November 2008 due to low solar elevation. For the Southern hemisphere sites Syowa and Neumayer Station data was available for January 1979 and November 2008.

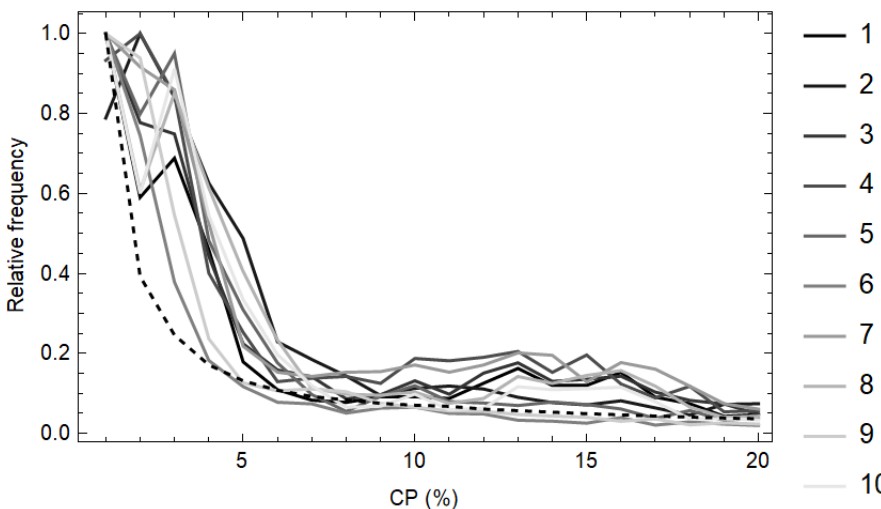

Figure 1. Ten largest pixelwise relative cloud probability distributions and the mean (dashed curve) of all *CP* distributions scaled with its maximum value. The solar zenith angle is restricted to not exceed 70°.

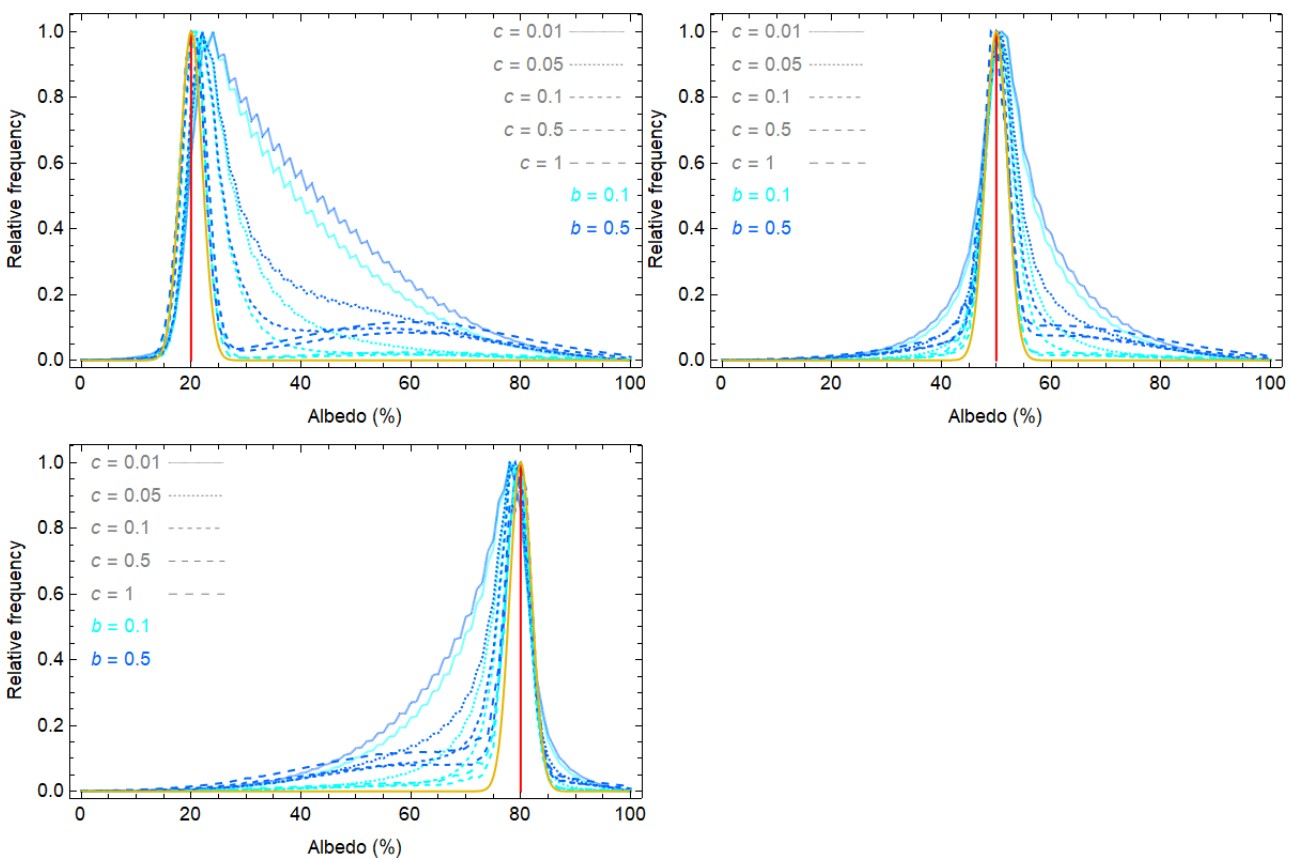

Figure 2. Examples of simulated cloudy albedo distributions for diverse values of parameters $b$ (blue shades) and $c$ (diverse values of dashing) in Eq. 5. The example surface albedo values are assumed to be Gaussian with mean values 20% (top left), 50% (top right) and 80% (top bottom) and a standard deviation of 2%. The surface albedo mean is shown as a red line and the distribution as a yellow curve. The parameter $p$ in Eq. 4 had a random value in the range [0,1]. The example cloud albedo
10  value is assumed to be Gaussian with a mean of 60% and standard deviation of 20%.





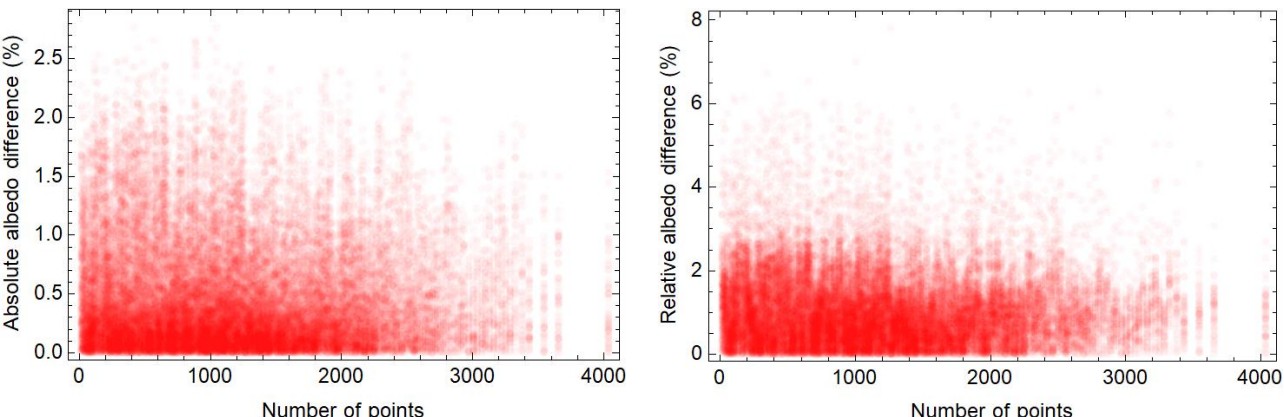

Figure 3. The relationship between the number of points in the cloud distribution and the simulated mean (left) and relative
mean (right) albedo accuracy.







Figure 4. The albedo retrieval distributions at Desert Rock, Payerne and Southern Great Plains in April 2009 and Syowa in November 2008.





5    Figure 5. The albedo retrieval distributions at Arctic Ocean (73.370°N, 139.180°W), Kara Sea (2.680°N, 62.860°E) and
Laptev Sea (75.320°N, 125.720°E) in April 2009 and July 2018. The cloud probability values of the individual black-sky
satellite based estimates are indicated by the colours. The monthly mean estimate is shown in red dashed line.





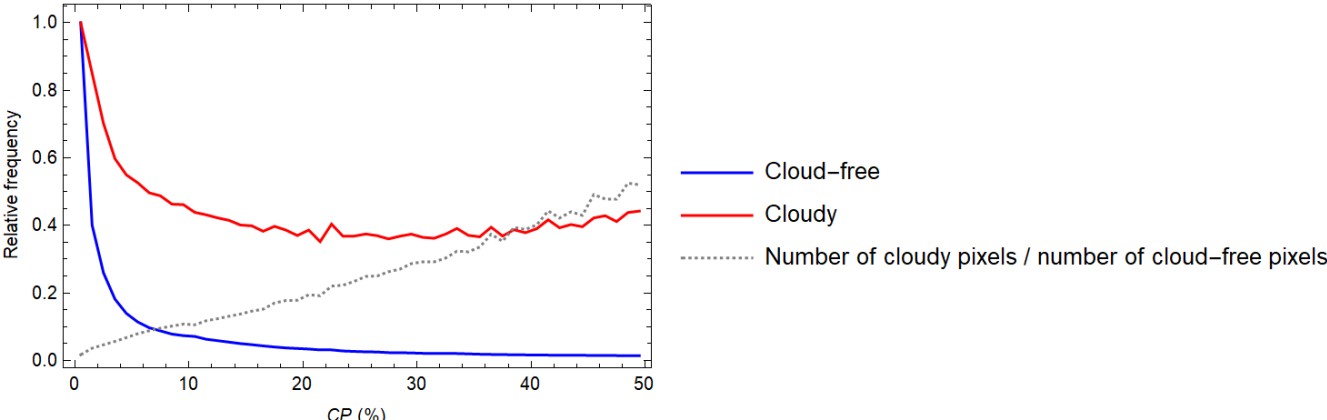

Figure 6. The relative distributions of cloud probabilities for pixels masked cloud-free and cloudy provided by the PPS
software for June 2012. The cloud-free masked pixel distribution is based on 26 million individual values and the cloudy
masked pixel distribution on 1.6 million individual values. The ratio of the number of pixels classified as cloud to the
number of pixels classified as cloud-free is shown as well.

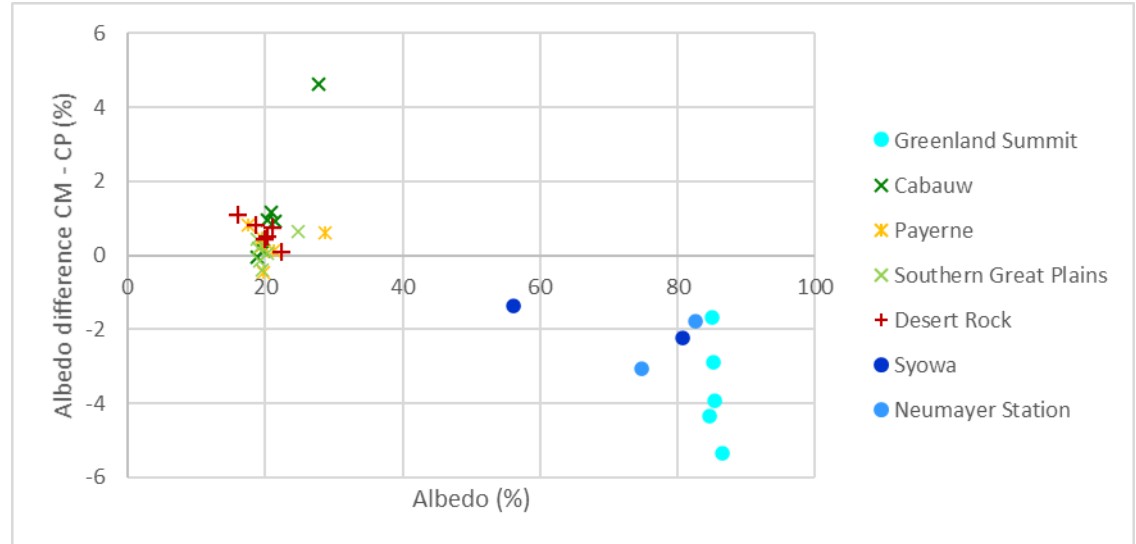

Figure 7. The difference between the monthly mean surface albedo estimates derived using the cloud masking (CM) and cloud probability (CP) approaches for several Northern hemisphere test sites in July 1979, April 1981, November 2008, April 2009, July 2018 and April 2020. For Greenland Summit no satellite albedo data was available for November 2008 due to low solar elevation. For the Southern hemisphere sites Syowa and Neumayer Station data was available for January 1979 and November 2008.