# Peer review of "Cloud probability -based estimation of black-sky surface albedo from AVHRR data"

_Atmospheric Measurement Techniques, 2021_

## Author Comment (AC1)

Answers to the comments of reviewer #1

In this paper, the authors attempt to determine surface albedo from AVHRR satellite measurements and with the help of cloud albedo distributions that replace a binary cloud masking approach.

The topic is clearly relevant to AMT and thus the venue is appropriate. As for the research, I confess that the article did not convince me either with respect to the novelty of the content, their representativeness or the analysis and conclusions.

The critical points that I found are: excessive use of concepts published in the past; assumptions too stringent regarding solar illumination and atmospheric state; a database too limited both in time (1 month, June 2012) and in space (only a few ground stations, without any comparison with other satellite datasets to appreciate the advantage of the CP inclusion).

1. The method to estimate the cloudy surface albedo is completely new, only the distribution formula is taken from the previous publication. In addition, the cloud probability data used in the simulations is completely new and also the station data used for validation. The basic equations describing the cloudy surface albedo are the ones published in 2004, but to enable a reader to understand the basics of the simulations, one has to explicitly write those equations (Eqs. 3-5). This is a common problem when using models that one is bound to repeat some text and equations already published to make the papers readable.

2. At the time of the work there was only one full month of cloud probability data ready, but it had global coverage. Hence, it represents cloudiness types of very large seasonal range from summer (north) to winter (south) and above diverse land cover types (forests, deserts, coastal areas etc.). Also, for the albedo there were at the development phase of the new albedo product data for a few test months, but they also represent a quite wide variety of land cover types and seasons (spring, summer, and winter). As station data represent the ground truth, it was essential to compare the derived satellite based albedo values with that, in order to check, whether the generalization of one month of global cloudiness is acceptable. The results support the view that the global one month of cloudiness is sufficiently representative subset of all year round global cloudiness to produce reliable simulation data in order to derive the parameter values to be used in calculating the surface albedo estimates. The new surface albedo product SAL times series using this method will naturally be compared with other satellite data, when the processing of the times series is finished, but that will be a separate study concentrating on other aspects than just tackling clouds. At this stage the new SAL retrieval method was compared with the previous method relying on binary cloud masking and the results supported choosing the cloudy probability based method presented in this manuscript. The previous version of the SAL product has been compared with the MODIS albedo product and the results were good.

The approach is also unclear to me. If you use only the CP of June 2012, how can you translate with confidence the method also for the months shown in Fig.7 and Tab.3?

Indeed, it was not known in advance, whether one month of global cloud probability data would be sufficiently representative, but the results support the view that the answer is

positive. I.e., the cloud probability based method produced albedo values closer to measured values than the binary cloud masking method.

In conclusion, the article still seems to me unrefined and not fully mature. It does not deliver a **compelling** message. Perhaps it would be useful to withdraw it, wait and rethink it not so much in the basic idea, which is valid, but in the development of the analysis.

So: to have more data available that would allow a deeper analysis and understanding of the variability that inevitably characterizes both the surface and the atmosphere.

The idea was to show fresh results that support the use of cloud probabilities instead of cloud masking. Processing of the full times series of the cloud probabilities and carrying out related simulations will be carried out when preparing the next release of the SAL product. Hence, publication of those results would take years. Waiting until that would unnecessarily slow down the development of other cloud probability based usage of satellite images. That's why the authors chose to write the manuscript now. Later on, the analysis will be repeated for the large data cloud probability data mass to update the coefficient values of Eq. 10, but that will be an effort within the CM SAF project not necessarily motivating writing a peer review paper.

I don't like to reject papers and I am conflicted about what judgment to give between major revisions and reject because on the one hand I would like the authors to have the opportunity to improve the work but on the other hand I find that the amount of improvements to be made is so substantial that it would be objectively easier to start over (personal opinion).

The discussion period of the journal should take only five weeks instead of five months and only the first review has now arrived. Due to the retirement of the first author within a month, it is neither possible to carry out a major revision nor a resubmission based on a lot of additional data/analysis. Hence, no publication will result from this work, unless the editor decides for a version based on the current material.

Main general comments:

1) I admit I was in trouble reading this paper because the part of the text from pages 5, line 28 to page 7 is a copy-paste of Manninen et al 2004. Although the similarity report gives a result of only 14%, it is surprising how the equations from 1 to 6 are the same, as well as the text with few variations. It is indeed work of the very same author, but I personally find the choice of copy-paste quite bold.

The text starting from page 5 line 28 is not a copy paste from the work carried out in 2004, it even contains a reference of year 2019 (page 5 line 33). The equations 1, 3, 4 and 5 are identical to those published in 2004, so that also the text describing them is close to that of the older publication. Indeed, it would be very odd, if the equations were *not* the same, as this work is a natural continuation to the work carried out in 2004. This is the problem with theoretical work: some basic equations have to be presented to enable the reader to understand the work carried out, but then one is bound to repeat text/equations that are already published. However, one should notice that only the starting point, i.e., the cloudy surface albedo distribution formula (Eq. 5, resulting from Eqs. 1-4), is the same here and in the previous publication, but the method how to derive the cloud-free albedo from that is

completely new in this manuscript (as well as all data, simulations, and analyses). The reason for that is that the previous method based on peak half-width and ¾-widths would have been much too slow for processing global time series of several decades. In addition, one has to take into account that in some areas the satellite based cloudy albedo distribution has only a rather small amount of individual albedo values. Therefore, a new robust method (Eqs. 7-12) was developed here. Furthermore, we added an explicit note to this section stating that the first part is built upon the preceding work from 2004 and reiterates the theoretical basis (page 7, lines 25-30). This should make it even more clear why some structural similarities are apparent.

This is not only a matter of form but also of substance: I am led to wonder where is the novelty in this research and the advancement in methods if the section "Theoretical cloud distributions" is taken from an article published in 2004 (17 years ago).

The novelty of this manuscript is not in the theoretical formulation of the cloudy surface albedo distribution, but in what follows: the new method to derive the cloud-free albedo from the cloudy albedo distribution and the use of the new cloud probability data to simulate realistic surface albedos in varying cloudiness, which are then validated with ground based measurements. The starting point of the theory was derived in 2004, but only now we have cloud probability values for each satellite image pixel and can derive empirical formulas for monthly means etc. Also, the validation data covers a larger variety here (summer and winter conditions). The new simulations, analysis and validation results give confidence in using satellite based cloud probabilities instead of traditional binary cloud masking.

Page 9 - Section 3.2 is also taken from Manninen et al 2004, Section 3.1 p 416, "Surface albedo algorithm". The same thing seems to me to apply when comparing Figure 5 of Manninen et al 2004 and Figure 2 of this paper.

This work deals with the surface albedo product SAL developed continuously in the CM SAF project of EUMETSAT since 1990's. The core of the current product is the same as in 2003, although there are also additions and updates made since that. Therefore, also the text of the first paragraphs of Section 3.2. describing the albedo product is very similar to that in the earlier publication. However, one should note that there are several references to newer publications than 2004, so that the text is really not a copy paste extract from the earlier publication.

The rest of Section 3.2. is completely new work that does not appear at all in the previous publication. It is based on using the new cloud probability data and the new method, how to derive the cloud-free albedo from the cloudy albedo.

The Figure 5 of the previous publication shows results from a theoretical simulation, whereas the Figure 2 of this manuscript shows a similar simulation results based on the empirical global cloud probability distribution used in this study.

I would like to genuinely ask the authors if they think there is enough scientific novelty in this AMT paper to justify its publication. Unlike the 2004 paper, they ingest cloud probability distributions but the results are still not dissimilar to the 2004 paper, as far as I understand.

The results of this manuscript are in line with the limited preliminary study of 2004 (three stations in Europe, summer conditions). However, the new method to derive the cloud-free

albedo values from the cloudy albedo distributions is worth publishing because it would not be possible to use the previous peak-widths based method for processing of decades of global data, as it would be far too time consuming. In addition, it is valuable to show that the results both in snow-free and snow-covered conditions are reasonably accurate in several BSRN stations in several continents.

I did check the similarity report too, and that 14% does not catch the semantics in my opinion. With some changes one can revamp old text in such a way to avoid a brute force database comparison, but conceptwise you are still sticking to old concepts. The authors seem to be aware of this and by citing every now and then the 2004 paper they avoided to write a much fairer sentence such as (e.g.) "From now on we apply the methodology developed in Manninen et al 2004." Period.

The flavour would be completely different. I honestly don't know how to deal with this situation.

Terhikki Manninen, Niilo Siljamo, Jani Poutiainen, Laurent Vuilleumier, Fred Bosveld, and Annegret Gratzki "Cloud statistics-based estimation of land surface albedo from AVHRR data", Proc. SPIE 5571, Remote Sensing of Clouds and the Atmosphere IX, (30 November 2004); https://doi.org/10.1117/12.565133

The authors could not have written "that a methodology developed in 2004 was used here", because that is not the case. The essential difference is that the cloud-free albedo values are derived from the cloudy distributions using a new method (Eqs. 7-12).

Quite a lot of scientific work is based on earlier theoretical work. As said before, the previously published equations 1-5 are the starting point for this work. From equation 7 onwards everything is new work. All Figures and Tables and Eqs. 7-12 are based on new work.

2) Unless I missed the information, other than the citation of the pyGAC package, the article makes no explicit mention of any corrections needed for AVHRR channel degradation, nor of the fact that the 40-year AVHRR record is composed of multiple platforms with different local overpass times, relevant for the task for **this** paper.

I imagine that both factors are relevant to the derivation of the surface albedo, both in all-sky configuration due to different atmospheres and black-sky albedo due to different illumination conditions (which I know the authors do not account for, but I am still puzzled by this choice).

This manuscript is not an overall description of all processing steps of the surface albedo product SAL, but is focused on how the cloudiness is dealt with. The starting point of this work is the top of atmosphere reflectance values of AVHRR, which are already intercalibrated. The original details of the intercalibration of the satellites and other quality control measures taken are described in Heidinger et al. (2010), as already cited in the text, with updates from the pyGAC development work cited in addition. This AVHRR data is based on the third revision of the original method by Heidinger et al., 2010. Unfortunately, there is not yet any new paper published on this latest revision, but in a revised version of this manuscript we can check if we can refer to the latest ATBD for PATMOS-x version 6 which

has recently been delivered to NOAA/NCEI. This PATMOS-x version also uses the latest revision of the calibration corrections.

Naturally, all possible defects of the TOA reflectance will enter in the final surface albedo value, but Table 3 shows that the achieved accuracy of the monthly mean albedo values is quite good. Obviously, the quality control steps preceding the TOA AVHRR reflectance retrieval have been successful.

Heidinger, A. K., Straka III, W. C., Molling, C. C., Sullivan, J. T., & Wu, X. (2010). Deriving an inter-sensor consistent calibration for the AVHRR solar reflectance data record. International Journal of Remote Sensing, 31(24), 6493-6517.

Trying to describe the multitude of details preceding the TOA reflectance retrieval would take far too much space and blur the focus of this manuscript. It is not possible to include every single detail in one publication, and it does not make sense either to copy a large number of details already published, when that information is not in the focus of the manuscript. Only the already published model is described in this manuscript, because that information is related to the core of the work done and the reader needs to understand it.

3) I was confused by the approach of the paper in that on the one hand it is described as a comprehensive study preparatory to reprocessing the CLARA dataset. On the other hand, however, very limited results are presented in terms of both atmospheric conditions and locations, with very stringent criteria on solar illumination and cloud type.

This manuscript describes how the cloudiness can be taken into account using cloud probabilities instead of binary cloud masking. All other processing steps are outside the scope of this work. They are described in the SAL ATBD. However, since the approach presented here is statistical, the exact details of the cloud types and solar illumination of individual cases are not required. The point of the simulations is to produce an extensive variation of cases and then derive a general method (Eqs. 7-12) that can be applied with reasonable accuracy to all cases. The reason for this strategy is that the surface albedo product has to be calculated also for the early years (1979 -> ), when very little information about the clouds or atmosphere exists. Hence, the albedo deriving method is developed to be robust. For that purpose, a large number of simulations was carried out.

Specific comments

- P2 L31: "with acceptable spatial representativeness of the site's measurement with respect to the albedo of the surrounding area".

It's not straightforward to me what this passage means. Or rather, I can guess that the authors want to make sure that the albedo around the measurement station does not vary drastically, so that a satellite overpass, that is not perfectly centered, is not contaminated by critically inhomogeneous surface types.

If my assumption is correct, I wonder if it is not useful instead to relax this criterion and analyze just what happens in very heterogeneous surface situations (e.g. coastal areas, mixed topography, urban settlements in arid areas, biologically active water masses).

I imagine the authors could agree that including the above cases would benefit the meaningfulness of their results.

Yes, the heterogeneity of the near surroundings of the measurement station would seriously complicate the comparison with the satellite based albedo, which has the spatial resolution of 5 km. On the basis of previous experience of albedo product validation, the authors do *not* agree that using heterogeneous stations would be beneficious. That would result in comparing apples with oranges in an uncontrollable way.

- P3 Section 2.2.2

I would like the authors to explain the reasoning behind the choice of the atmospheric correction approach of Rahman and Dedieu and the selection and filtering criteria of AOD.

AI is an index and is it still differentially sensitive to so many aerosol properties and line-of-sights that is interesting (or misterious) to me how it can be used for this task.

The atmospheric correction approach of the SAL product has been based on the SMAC algorithm because of its computational efficiency. Taking into account that the albedo time series is global and covers several decades, more complicated approaches have so far not been realistic for the full times series. Another challenge for the atmospheric correction is that there is no global AOD data available for the earliest decades (starting 1979-). The only aerosol related global information of that time is the aerosol index based on TOMS (and later on OMI) data. For time series analysis homogeneity of the data set is important, hence the AOD is derived similarly for the whole time series, although in the latter years other options could have been used as well. However, the AOD time series optimized for atmospheric correction by Jääskeläinen et al. (2017) has been shown to compare well with atmospheric correction carried out using several satellite based AOD products (see https://www.mdpi.com/2072-4292/9/11/1095/htm).

P4 L14 : Figure 1 can be greatly improved. I personally would not cut it at 20% but leave the full X-axis domain and the 20% subset as inset. Also in view of the discussion in the next paragraph about the U-shaped distribution. There (P5 L12) Figure 1 is invoked but the U-shaped distribution is not intuitable.

In a revised version, this suggestion will be taken into account.

In the ensuing text also it appears to be introduced as a synthesis of AVHRR data given at native resolution 1.1 km and the GAC product (5 km). Information that is not given in the caption of the figure.

There appears to have been a misunderstanding here regarding the use of the word "pixel", which the reviewer seems to have understood as referring to the original LAC (1.1km) resolution imagery. All AVHRR data used in the manuscript are from the GAC resolution data records – the text is now revised to make this clear.

- P5 L4-6: "When estimating the cloud fraction distribution over the entire globe in a very coarse spatial resolution, however, it is possible that the extreme values are not achieved at all."

I disagree with this statement. On the one hand, Krijger et al ( https://doi.org/10.5194/acp-7-2881-2007 ) have shown that even at the spatial resolution of GOME (320 x 40 km2) - which is to my knowledge the sensor with the coarsest spatial resolution used in cloud remote sensing - there is a non-negligible probability of having cloud-free pixels. Speaking of the other extreme, CF = 1, we know well that there are synoptic-scale (~1000 km) cloud systems that can be fully covered by the swath of such a sensor.

There are numerous studies comparing CF from GOME with real data and it is clear that the U-distribution of cloud fraction is largely (not completely) independent of the spatial resolution of the instrument. What makes the difference is the algorithm and the class of clouds under consideration.

The first two that come to my mind.

Lutz, R., Loyola, D., Gimeno García, S., and Romahn, F.: OCRA radiometric cloud fractions for GOME-2 on MetOp-A/B, Atmos. Meas. Tech., 9, 2357–2379, https://doi.org/10.5194/amt-9-2357-2016, 2016.

Grzegorski, M., Wenig, M., Platt, U., Stammes, P., Fournier, N., and Wagner, T.: The Heidelberg iterative cloud retrieval utilities (HICRU) and its application to GOME data, Atmos. Chem. Phys., 6, 4461–4476, https://doi.org/10.5194/acp-6-4461-2006, 2006.

So if the authors mean the native resolution of an instrument at the ground (footprint), in my opinion, they are wrong. Alternatively, one could talk about gridded cloud fraction resolution. Perhaps after aggregation with arbitrary temporal and spatial sampling the extremes will never be reached. I invite the authors to reconsider the logic of their reasoning.

The authors had not any spatial resolution of a special instrument in mind, when writing this sentence. It is meant more theoretically. The coarser the resolution is, the lower is the probability for completely cloud-free pixels. The utmost limit is then just one pixel for the whole globe, and for that the value is never zero.

P6 L7-9: "The cloud albedo distribution can also be assumed Gaussian, although the standard deviation may be so large, that the result is essentially the same as for uniform distribution."

This is a surprising and simplifiying statement. The albedo of clouds is primarily a function of their optical thickness, which is never normally distributed. It has been shown that the albedo of clouds is better approximated by a beta and Weibull distribution (i.e. Koren and Joseph, 2000).

Koren, Ilan, and Joachim H. Joseph. "The histogram of the brightness distribution of clouds in high― resolution remotely sensed images." Journal of Geophysical Research: Atmospheres 105.D24 (2000): 29369-29377.

The comment is important. When one considers the cloud albedo of a certain cloud type, the distribution is better approximated by a beta and Weibull distributions, as the paper referred to shows. However, here it is not a question of the albedo of a certain cloud type, but the albedo of all clouds within a pixel. Moreover, the pixels are so coarse (5 km) compared to the study by Koren and Joseph (50 m) that they may have several cloud types and more than one cloud layer within the pixel. Hence, one can't expect that a distribution of a single cloud type

would be representative for this coarse pixels. Then the Gaussian distribution is a simple choice, as random sampling from the bell shaped distributions of diverse cloud types would come very close to that, as even the original shapes in the paper by Koren and Joseph do not deviate much from the Gaussian. The Gaussian distribution was also supported by the study made in 2004 in 1.1 km resolution using ceilometer data.

However, to clarify this issue, the above reference and a comment related to relation of the distribution and spatial and temporal resolution will be added to a revised version of the manuscript.

P 11 L 11-12: "The difference increases with increasing AOD". Could you expand this sentence and give more information about the AOD values, how they are measured, and the type of aerosol?

The AOD values used are described in Jääskeläinen et al. (2017, https://www.mdpi.com/2072-4292/9/11/1095/htm). As already mentioned for the earliest years of the SAL product there are no aerosol measurements available, except the aerosol index derived from TOMS. The details, how that is data is used for deriving the AOD input for the atmospheric correction of SAL are given in Jääskeläinen et al., 2017 (https://www.mdpi.com/2072-4292/9/11/1095/htm). The sentence will be expanded in a revised version of the manuscript.

P12 L 13: "The chosen limit CP < 20% is a compromise between the quality of TOA reflectance values and the number of pixels available for a monthly mean albedo retrieval"

What does that "quality of TOA reflectance" mean? Can you give figures of the radiometric accuracy needed to achieve the results you are presenting? I am convinced that this is important information, since we are talking about a satellite product that should be used as input for other algorithms.

Here the quality of the TOA reflectance value refers to the cloud probability. If one restricted the analysis to pixels with CP = 0, the number of pixels would be really small and statistically thus not that reliable. On the other hand, the larger CP value is included in the analysis, the lower will be the quality of the derived cloud-free surface albedo value, as the analysis will then be dominated by cloudy pixels.

The radiometric quality of the TOA reflectance values is observed while processing the cloud probabilities and all dubious pixels (for example those having sun glints) are discarded from further analysis. The remaining question is how to derive cloud-free surface albedo from the TOA reflectance values that have already passed the quality checks. Binary cloud masking or using cloud probabilities? Both approaches have their pros and cons, but Figure 7 and Table 3 support the use of CP values instead of the binary cloud mask.

As the radiometric accuracy of AVHRR GAC data is inferior to that of modern optical satellite instruments, such as MODIS or Sentinel-2, the successful use of the cloud probability values instead of cloud masking for GAC data supports the use of the method for data obtained from instruments with better radiometric accuracy than old AVHRRs.

P12 L30: "In addition, the difference between the estimates of the two methods is typically largest for snow-covered areas, where cloud discrimination is very challenging, especially when the sun elevation is low".

I don't understand then the sense of this study, if you are not able to separate and isolate the factors that contribute to the differences in the albedo. The authors rely on this argument several times in the text, but I wonder why they couldn't just look for an RGB image from a high-resolution satellite to show that there really is heterogeneous and patchy snow cover, for instance.

It seems that the reviewer misunderstood the sentence. The authors emphasized that the largest difference between binary cloud masking and the cloud probability based method takes place in the most difficult conditions, i.e., in snow-covered areas, where the sun elevation is low. This means that the major improvement due to the new method is obtained in the most difficult areas. Is that not a result worth striving for?

P13 L 6: "The CLARA-A3 SAL will be derived using the CP values instead of the binary cloud mask. The pentad means will be derived technically similarly as the monthly means using pentad distributions of CP."

What is the "pentad" distribution? Why does it need to be introduced here in the discussion of results without any context?

A pentad distribution is a distribution of five days instead of one month. As both monthly and pentad means are provided in the SAL product, it is mentioned here that the pentads follow the same logic as the monthly values.

P13 L 7: "Future studies of the CLARA-A3 CP and cloud mask characteristics will show, whether it would be desirable to use both the cloud mask and the CP values as the basis for SAL estimation."

I thought the purpose of this study was really to show that using CP distributions was advantageous over using a CM approach. However, here in the conclusion it says that it has not yet been decided. This statement leads me to think that even the authors themselves are aware of the limited informative value of this study.

Yes, this study showed that using CP values instead of binary cloud mask improves the albedo retrieval accuracy. It is already decided that the next SAL version is derived using the CP values instead of the binary cloud mask. However, the authors did not want to exclude the possibility of using both a binary cloud mask and the CP values in the future. Perhaps in connection with machine learning algorithms etc. The strategy of the SAF products by EUMETSAT is continuous operations and development, which means that methods being used now may be surpassed with better methods later.

P23 Table 3: No statistics of differences are given for the sites.

As the original values are given, differences would be redundant, but they can easily be added to a revised version of the manuscript.

Minor comments

- P2 Last paragraph of the introduction. I personally am a proponent of a description of the structure of a paper at the end of the introductory section (e.g. in section 2 the data are introduced, while in 3 and 4 the reader finds ... )

In a revised version this comment will be taken into account.

- P3 L8: what does the acronym FDR mean? As a section title, expand it.

FDR is fundamental data record as mentioned in line 16. The section title will be changed to Fundamental data record (FDR).

Typos

- P5 L 10: "Although the cloud probability estimation is complicated various kinds of uncertainties" -> by (?)

- P8 L12 : than -> then

The typos will be taken into account in a revised version.

---

## Author Comment (AC2)

Answers to the comments of reviewer #2 are in blue.

**General Comments**
This paper presents a statistical method of calculating temporally averaged black-sky surface albedo from measurements made by a satellite imaging radiometer - in this case AVHRR. The unique aspect of the method presented is that it includes measurements effected by partial cloud-cover, using a cloud-probability (CP) product (essentially the Bayesian probability that a given observed pixel is, or is not, cloudy) to correct the albedo derived from top-of-atmosphere observations with a given CP threshold. The method is presented as an improvement on previous albedo retrieval schemes which rely on binary cloudy-clear masks. The authors provide a derivation of the equations used to make this correction, with a description of the assumptions and limitations of the method, before presenting results of the algorithm applied over a small range of stations which provide in-situ surface albedo observations.

The work presented is interesting, especially as the method is being operationally applied to calculate surface albedo in the new CLARA-A3 AVHRR products produced by the CM-SAF, and the derivation and analysis seem sound. The paper draws heavily on work done previously by the lead author (Manninen et al. 2004) and represents the (long-awaited, one imagines) practical realisation of that more theoretical analysis. Thus, as an improvement and application of an existing approach, which is being applied to a large data record, I feel it is worthy of publication. However, the paper itself could do with some improvement. My biggest complaint is the paper lacks a clear description of its structure - there is a brief (3 sentence) overview of what the paper covers, but without an existing knowledge of the analysis undertaken by the authors, I felt lost for much of the paper. The authors have a tendency to provide a series of related, but not directly connected statements, which makes following the thread challenging. Thus, I would recommend that the introduction is extended, or an introductory section is added to the methods (section 3), to include a overview of the algorithm which clearly lays-out the steps involved and the final product - maybe a flow diagram would help.

Thank you for encouragement. The manuscript will be revised by adding to the end of the introduction the following paragraph:
The in situ albedo data used for validation of the satellite based albedo estimates are presented in Section 2.1. The satellite data used is described in Section 2.2 with emphasis on the atmospheric correction (Section 2.2.2) and the cloud probabilities (Section 2.2.3). The method how to take cloudiness into account when estimating the surface albedo is described in Section 3. The essential points of a previous theoretical study (Manninen et al., 2004) of deriving cloudy albedo distributions are summarized in Section 3.1.1. Then the approach is further developed to adapt it to cloudy surface albedo simulations based on the cloud probability data (Section 3.1.2) and finally a new method how to derive the cloud-free surface albedo using cloud probabilities is presented (Section 3.2).

The method section will be provided the following introduction:
The cloud-free surface albedo estimates of CLARA-A3 will be estimated using the TOA reflectance and CP values available in pixel basis (Figure 1). First the TOA reflectance values with CP > 20% are discarded, as well as values flagged as low quality by the PPS software, for example because of sun glints. Then the atmospheric correction is carried out the same way for all remaining TOA reflectances independently of the cloud probability. Finally, the

monthly mean cloud-free surface albedo is estimated using the atmospherically corrected reflectances and corresponding CP values. The main points of the theoretical background for the cloudy surface albedo distributions (Manninen et al., 2004) are summarized in Section 3.1.1. The adaptation of the theoretical approach to using cloud probability data is described in Section 3.1.2 and finally the formulas for deriving the cloud-free monthly mean surface albedo estimates are provided in Section 3.2.

The flow diagram describing how the cloudiness is taken into account when estimating monthly mean surface albedo values will be a new Figure 1:

[Figure]

One specific omission in the paper is that no indication of which wavelength(s) the albedo is being derived for. I presume it is one or more of the AVHRR visible/near-IR bands. Please include this information in the paper.

Yes, the visible and near infrared bands are used. This will be made clear in the revised manuscript.

**Specific corrections and suggestions**
Abstract: The abstract doesn't scan well and should be revised. For example the basic purpose of the paper should be stated in the very first sentence, so the abstract should start will something like (as an example): "This paper describes a new method for cloud-correcting observations of black-sky surface albedo derived using the Advanced Very High Resolution Radiometer (AVHRR)."

The abstract will be revised as suggested:
”This paper describes a new method for cloud-correcting observations of black-sky surface albedo derived using the Advanced Very High Resolution Radiometer (AVHRR). Cloud cover constitutes a major challenge for the surface albedo estimation using AVHRR data for all possible conditions of cloud fraction and cloud type on any land cover type and solar zenith angle. This study shows how the new cloud probability (CP) data to be provided as part of the edition A3 of the CLARA (CM SAF cLoud, Albedo and surface Radiation dataset from AVHRR data) record by the project Satellite Application Facility on Climate Monitoring (CM SAF) of EUMETSAT can be used instead of traditional binary cloud masking to derive cloud-free monthly mean surface albedo estimates. Cloudy broadband albedo distributions were simulated first for theoretical cloud distributions and then using global cloud probability (CP) data of one month. A weighted mean approach based on the CP

values was shown to produce very high accuracy black-sky surface albedo estimates for simulated data. The 90% quantile for the error was 1.1% (in absolute albedo percentage) and for the relative error it was 2.2%. AVHRR based and in situ albedo distributions were in line with each other and also the monthly mean values were consistent. Comparison with binary cloud masking indicated that the developed method improves cloud contamination removal."

Pg.1, Ln.20: Again, these introductory sentences don't scan well and come across as a series of dis-connected sentences. For example, I would suggest re-structuring the first few sentences like so: "The surface albedo is a key indicator of climate change (GCOS, 2016) and is continuously and accurately measured across contrasting climatic zones by the Baseline Surface Radiation Network (BSRN), operated by the World Climate Research Programme (WCRP). However, satellite remotes sensing is required to augment these regional measurements with global estimates of surface albedo".

The beginning of the introduction will be revised as suggested.

Pg.2, Ln.11: I'm not sure what is meant by the sentence "However, for the really large deviations also other cloudy vs clear non-separability issues become important"

This sentence and the previous ones in the text discuss the cases when cloud detection fails and falsely labels a pixel as being cloud-free. This concerns mainly very thin clouds and is problematic for the surface albedo retrieval if radiances in the two visible and near-infrared channels are then still higher than what the surface would produce in the true cloud-free case. However, there are situations for high solar zenith angles when low-level clouds are missed even if they are optically thick (e.g. fog or stratus). These clouds may give near-zero reflectances despite being optically thick, typically if these clouds are shadowed by other clouds or mountains. The reason for not being detected in these cases is typically that cloud top temperatures are close to surface temperatures meaning that not even in the infrared AVHRR channels there is a typical cloud signature (clouds are normally colder than the surface). The impact on surface albedo retrievals for such a case, which might be quite serious over snow-covered surfaces, depends on the maximum allowed solar zenith angle that is used for the surface albedo retrieval. If this threshold is too close to 90 degrees, the risk to encounter shadowed mis-classified clouds might be high. As a consequence, surface albedo retrievals might give unrealistic visible and NIR reflectances coming from clouds rather than from the underlying surface. For a snow-covered surface this might lead to an underestimated surface albedo. Hence, the CLARA surface albedo product is limited to cases, when the solar zenith angle is $\leq 70°$.

The following revised text is proposed:

" However, for the really large deviations also other cloudy vs clear non-separability issues become important. For example, low-level clouds being in shadow at high solar zenith angles (e.g., caused by higher level clouds or mountain peaks) might be missed as a consequence of having non-typical visible and NIR reflectances as well as a lacking temperature difference between the cloud top and the surface. If such missed clouds occur over snow-covered surfaces they might lead to a seriously underestimated surface albedo. Using such data would introduce errors on the order of 100% on derived surface albedo, with potentially much higher errors occurring in cases with the combination of  snow, complex terrain and low sun elevation, which are common in Northern Europe for example. For this reason, the surface

albedo of the CLARA surface albedo product is restricted to limited to cases, for which the solar zenith angle ≤ 70°."

Pg.2, Ln.13-15: I would suggest replacing the last two sentences of this paragraph is something more succinct. For example: "Using such data would introduce errors on the order of 100% on derived surface albedo, with potentially much higher errors occurring in cases with the combination of  snow, complex terrain and low sun elevation, which are common in Northern Europe for example."

The text will be edited as suggested.

Pg.2, Ln.19-21: A couple of points here. Firstly, the sentence needs restructuring, I would suggest something like: "Thus, across a 0.25 x 0.25 degree grid-box over one month, the slowly varying surface albedo would be expected to dominate the broadband albedo distribution observed by non-cloud masked AVHRR data". The second question is, why would you expect the albedo distribution to be dominated by the surface contribution, even though the cloud albedo is more variable? Surely this would be rather dependent on how much, and just how variable, the cloud cover was for the region and period in question?

The text will be edited as suggested.

The total distribution of a cloudy region albedo can be thought to be a combination of the cloud albedo distribution and the cloud-free surface albedo distribution. When there are equal number of both cases the distribution of the less varying target has the higher peak, i.e. it dominates the total distribution from the point of view of the highest peak. Below are three fictive case demonstrations for that (top left, top right and bottom left). It is true that the variability of the cloud cover affects also the total distribution. However, the highest peak comes from the least varying target, typically the surface. But even if the number of cases is larger for the broader distribution (below lower right figure), the narrower distribution peak is usually still higher.

[Figure]

Pg.2, Ln.32: Replace "surrounding area, an important" with "surrounding area, which is an important".

The text will be edited as suggested.

Section 3.1.1 I feel this section would benefit from restructuring. As it stands, it reads like a series of seemly unconnected statements. For example, Pg.5 starts with a description of the distribution of cloud fraction and then suddenly switches to the diurnal variation of surface black-sky albedo, before switching again to seasonal and monthly variation of surface albedo. A simple introductory statement laying out what albedo components are to be discussed and why at the start of the section is required - something along the lines of what appears starting at Pg.6, Ln.5, for example.

This section will be revised as suggested.

Pg.5, Ln.9: Replace "like ceilometer observations show" with "as is shown by ceilometer observations, for example"

The text will be edited as suggested.

Pg.5, Ln.12: I'm not sure how Figure 1 could be described as resembling a U-curve. If this is not an error, more explanation is needed.

Figure 1 shows the left part of the U-curve. As almost 100% cloudy pixels will not be suitable for albedo estimation, it is not considered of interest, what the right part of the U-curve looks like, as it will never be used for albedo retrieval. The question is what is the reasonable CP threshold: 50% or smaller. Obviously the text was written unclearly and will be revised to clarify the issue.

Pg.5, Ln.17/18: Remove "also".

The text will be edited as requested.

Pg.11, Ln.7: Remove comma after "shown".

The text will be edited as requested.

Pg.11, Ln.27: "high" rather than "highest".

The text will be edited as requested.

Pg.12, Ln.1: Replace "zenith angle so that" with "zenith angle such that".

The text will be edited as requested.

Pg.12, Ln.7: Remove "per pass".

The text will be edited as requested.

Pg.12, Ln.10: "also provides" rather than "provides also".

The text will be edited as requested.

Pg.13, Ln.9: Remove comma after "show".

The text will be edited as requested.

Figure.3: These plots do not effectively convey the distribution of the points plotted, beyond showing they are concentrated in the bottom left corner. I would suggest a density plot (where the data-space is divided into a regular grid and the number of points in each bin is shown by a colour gradient).

The figures are edited as suggested.

[Figure]

Figure.4: I assume the top-left panel should be labelled "Desert Rock", rather than "Payerne"? Also, I don't think it is necessary to show the full range of albedo for each panel - the distributions would be clearer if the x-axis was limited to the range of albedo observed at each station.

It seems the Payerne figure was erroneusly provided twice and the Desert Rock Figure was missing. It should have been this one:

[Figure]

This will be corrected and the scales will be adjusted as suggested.

Figure.5: See figure.4.

The scales will be adjusted as suggested.

Figure.6: I would suggest that this plot be regenerated to show the distributions of CP values flagged as cloudy or clear relative to the total number of observations of at each CP value (so that the sum of the red and blue lines is always 1). This would convey the the distributions in a more intuitive way and remove the need to include the dotted "cloud-fraction" line.

The Figure will be revised as suggested: